# MDS Self-Dual Codes and Antiorthogonal Matrices over Galois Rings

**Sunghyu Han** 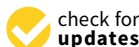

School of Liberal Arts, KoreaTech, Cheonan 31253, Korea; sunghyu@koreatech.ac.kr

**Abstract:** In this study, we explore maximum distance separable (MDS) self-dual codes over Galois rings $GR(p^m, r)$ with $p \equiv -1 \pmod 4$ and odd $r$. Using the building-up construction, we construct MDS self-dual codes of length four and eight over $GR(p^m, 3)$ with ($p = 3$ and $m = 2, 3, 4, 5, 6$), ($p = 7$ and $m = 2, 3$), ($p = 11$ and $m = 2$), ($p = 19$ and $m = 2$), ($p = 23$ and $m = 2$), and ($p = 31$ and $m = 2$). In the building-up construction, it is important to determine the existence of a square matrix $U$ such that $UU^T = -I$, which is called an antiorthogonal matrix. We prove that there is no $2 \times 2$ antiorthogonal matrix over $GR(2^m, r)$ with $m \geq 2$ and odd $r$.

**Keywords:** antiorthogonal matrices; Galois rings; MDS codes; self-dual codes

**MSC:** 94B05

## 1. Introduction

In coding theory, the minimum distance is very important because it indicates the ability to perform error correction on the codes. Therefore, maximum distance separable (MDS) codes have attracted much attention. However, self-dual codes have also been investigated because they are closely related to other mathematical structures such as block designs, lattices, modular forms, and sphere packings (see [1] as an example). Codes that contain both structures, which are called MDS self-dual codes, have been investigated. MDS self-dual codes have been studied over finite fields (see [2] as an example) and over finite rings $\mathbb{Z}_{p^m}$ (see [3] as an example). They have also been studied for Galois rings $GR(p^m, r)$. For the $p = 2$ case, the codes were investigated in [4], and for $p \equiv 1 \pmod 4$ with any $r$ or $p \equiv -1 \pmod 4$ with even $r$, the codes were studied in [5]. To the best of our knowledge, MDS self-dual codes have not yet been studied for the case $p \equiv -1 \pmod 4$ with odd $r$, and this case is the focus of our study. Specifically, we constructed several MDS self-dual codes over $GR(p^m, 3)$.

There are several construction methods for self-dual codes. One of the methods is the building-up construction. The method was first described in [6] and then developed in many papers (see [7] as an example). We note that the method was considered for $GR(p^m, r)$ with $p \equiv -1 \pmod 4$ and odd $r$ in [8]. In this paper, we used the building-up construction method which was described in [9] to construct MDS self-dual codes. For this method, it is very important to determine the existence of a square matrix $U$ such that $UU^T = -I$, which is called an antiorthogonal matrix [10,11]. There has been a study aimed at dealing with this problem [12]. In this study, we performed additional research on this problem.

This paper is organized as follows. In Section 2, we present basic facts regarding Galois rings, linear codes over $GR(p^m, r)$, MDS codes, self-dual codes, and building-up construction. In Section 3, we discuss MDS self-dual codes over $GR(p^m, r)$ with $p \equiv -1 \pmod 4$ and odd $r$, and we give our computational



results for those codes. Specifically, we construct MDS self-dual codes of length four and eight over $GR(p^m, 3)$ with ($p = 3$ and $m = 2, 3, 4, 5, 6$), ($p = 7$ and $m = 2, 3$), ($p = 11$ and $m = 2$), ($p = 19$ and $m = 2$), ($p = 23$ and $m = 2$), and ($p = 31$ and $m = 2$). In Section 4, we review previously reported results showing the existence of antiorthogonal matrices, and we prove that there is no $2 \times 2$ antiorthogonal matrix over $GR(2^m, r)$ with $m \geq 2$ and odd $r$. All of the computations in this paper were performed using the computer algebra system Magma [13].

## 2. Preliminaries

In this section, we present basic facts regarding Galois rings, linear codes over Galois rings $GR(p^m, r)$, MDS codes, self-dual codes, and building-up construction. We start with Galois rings.

### 2.1. Galois Rings

In this subsection, we present some well-known facts about Galois rings (see [14] as an example). Let $p$ be a fixed prime and $m$ be a positive integer. First, we consider the following canonical projection

$$\mu : \mathbb{Z}_{p^m} \to \mathbb{Z}_p \tag{1}$$

which is defined by

$$\mu(c) = c \pmod{p}. \tag{2}$$

The map $\mu$ can be extended naturally to the following map

$$\mu : \mathbb{Z}_{p^m}[x] \to \mathbb{Z}_p[x] \tag{3}$$

which is defined by

$$\mu(a_0 x + a_1 x + \cdots + a_n x^n) = \mu(a_0) + \mu(a_1)x + \cdots + \mu(a_n)x^n. \tag{4}$$

This extended $\mu$ is a ring homomorphism with kernel $(p)$.

Let $f(x)$ be a polynomial in $\mathbb{Z}_{p^m}[x]$. Then, $f(x)$ is called basic irreducible if $\mu(f(x))$ is irreducible. A Galois ring is constructed as

$$GR(p^m, r) = \mathbb{Z}_{p^m}[x]/(f(x)), \tag{5}$$

where $f(x)$ is a monic basic irreducible polynomial in $\mathbb{Z}_{p^m}[x]$ of degree $r$. The elements of $GR(p^m, r)$ are residue classes of the form

$$a_0 + a_1 x + \cdots + a_{r-1} x^{r-1} + (f(x)), \tag{6}$$

where $a_i \in \mathbb{Z}_{p^m}, (0 \leq i \leq r - 1)$. The ring homomorphism $\mu$ induces a ring homomorphism $\overline{\mu}$

$$\overline{\mu} : GR(p^m, r) = \mathbb{Z}_{p^m}[x]/(f(x)) \to \mathbb{F}_{p^r} = \mathbb{Z}_p[x]/(f(x)) \tag{7}$$

which is defined by

$$\overline{\mu}(g(x) + (f(x))) = \mu(g(x)) + (\mu(f(x))). \tag{8}$$

A polynomial $h(x)$ in $\mathbb{Z}_{p^m}[x]$ is called a basic primitive polynomial if $\mu(h(x))$ is a primitive polynomial. It is a well-known fact that there is a monic basic primitive polynomial $h(x)$ of degree $m$ over $\mathbb{Z}_{p^m}$ and $h(x)|(x^{p^r-1} - 1)$ in $\mathbb{Z}_{p^m}[x]$. Let $h(x)$ be a monic basic primitive polynomial in $\mathbb{Z}_{p^m}[x]$ of degree $r$. Consider the following element

$$\xi = x + (h(x)) \in GR(p^m, r) = \mathbb{Z}_{p^m}[x]/(h(x)). \tag{9}$$

Then, the order of $\xi$ is $p^r - 1$. Teichmüller representatives are defined as follows.

$$T = \{0, 1, \xi, \xi^2, \ldots, \xi^{p^r-2}\}. \tag{10}$$

Then, every element $t \in GR(p^m, r)$ can be uniquely represented by the form

$$t = t_0 + pt_1 + p^2 t_2 + \cdots + p^{m-1} t_{m-1}, \tag{11}$$

where $t_i \in T, (0 \leq i \leq m-1)$. Using this notation, we define the following map $\sigma$

$$\sigma : GR(p^m, r) \to GR(p^m, r) \tag{12}$$

by

$$\sigma(t) = t_0^p + pt_1^p + p^2 t_2^p + \cdots + p^{m-1} t_{m-1}^p. \tag{13}$$

The following facts are known.

1. $\sigma$ is a ring automorphism of $GR(p^m, r)$.
2. $\sigma$ fixes every element of $\mathbb{Z}_{p^m}$.
3. $\sigma$ is of order $r$ and generates the cyclic Galois group of $GR(p^m, r)$ over $\mathbb{Z}_{p^m}$.

### 2.2. Linear Codes over $GR(p^m, r)$

A linear code $C$ of length $n$ over $GR(p^m, r)$ is a submodule of $GR(p^m, r)^n$, and the elements in $C$ are called codewords. The distance $d(\mathbf{u}, \mathbf{v})$ between two elements $\mathbf{u}, \mathbf{v} \in GR(p^m, r)^n$ is the number of coordinates in which $\mathbf{u}, \mathbf{v}$ differ. The minimum distance of a code $C$ is the smallest distance between distinct codewords. The weight of a codeword $\mathbf{c} = (c_1, c_2, \cdots, c_n)$ in $C$ is the number of nonzero $c_j$. The minimum weight of $C$ is the smallest nonzero weight of any codeword in $C$. If $C$ is a linear code, then the minimum distance and the minimum weight are the same.

A generator matrix for a linear code $C$ over $GR(p^m, r)$ is permutation equivalent to the following one in the standard form [15,16]:

$$G = \begin{pmatrix} I_{k_0} & A_{0,1} & A_{0,2} & A_{0,3} & \cdots & A_{0,m-1} & A_{0,m} \\ 0 & pI_{k_1} & pA_{1,2} & pA_{1,3} & \cdots & pA_{1,m-1} & pA_{1,m} \\ 0 & 0 & p^2 I_{k_2} & p^2 A_{2,3} & \cdots & p^2 A_{2,m-1} & p^2 A_{2,m} \\ \vdots & \vdots & \vdots & \vdots & & \vdots & \vdots \\ 0 & 0 & 0 & 0 & \cdots & p^{m-1} I_{k_{m-1}} & p^{m-1} A_{m-1,m} \end{pmatrix}, \tag{14}$$

where the columns are grouped into square blocks of sizes $k_0, k_1, \ldots, k_{m-1}$. The rank of $C$, denoted by $\text{rank}(C)$, is defined to be the number of nonzero rows of its generator matrix $G$ in a standard form. Therefore $\text{rank}(C) = \sum_{i=0}^{m-1} k_i$. We call $k_0$ in $G$ the free rank of a code $C$. If $\text{rank}(C) = k_0$, then $C$ is called a free code. We say $C$ is an $[n, k, d]$ linear code, if the code length is $n$, the rank of $C$ is $k$, and the minimum weight of $C$ is $d$. In this paper, we assume that all codes are linear unless we state otherwise.

### 2.3. MDS Codes

It is known (see [17] as an example) that for a (linear or nonlinear) code $C$ of length $n$ over any finite alphabet $A$,

$$d \leq n - \log_{|A|}(|C|) + 1. \tag{15}$$

Codes meeting this bound are called MDS codes. Further, if $C$ is a linear code over a ring, then

$$d \leq n - \text{rank}(C) + 1. \tag{16}$$

Codes meeting this bound are called maximum distance with respect to rank (MDR) codes [16,18]. The presence of MDR codes does not imply MDS codes. See the following example.

**Example 1.** *Let C be a linear code generated by $G = (2)$ over $\mathbb{Z}_4$. Then, $n = 1$, $\text{rank}(C) = 1$, and $d = 1$. Therefore, C is an MDR code. Because $\log_{|A|}(|C|) = \log_4 2 = \frac{1}{2}$, C is not an MDS code.*

The following lemma states the necessary and sufficient condition for MDS codes.

**Lemma 1.** *A linear code C is MDS if and only if C is MDR and free.*

**Proof.** ($\Rightarrow$) If $C$ is not free, then $\log_{|A|}(|C|) < \text{rank}(C)$. Therefore, $d \leq n - rank(C) + 1 < n - \log_{|A|}(|C|) + 1$, so $C$ should be free, and $\log_{|A|}(|C|) = \text{rank}(C)$. Thus, $C$ is MDR.
($\Leftarrow$) Let $\text{rank}(C) = k$. Then, $|C| = (p^{mr})^k$. Because $|A| = p^{mr}$, we have $\log_{|A|}(|C|) = k = \text{rank}(C)$. Therefore, $C$ is MDS. $\square$

The following theorem states that the weight distribution of MDS codes over $GR(p^m, r)$ of code length $n$ is uniquely determined.

**Theorem 1** ([16] Theorem 5.10). *Let C be an MDS code over $GR(p^m, r)$ of code length n and minimum weight d. For $d \leq w \leq n$, denote by $A_w$ the number of words of weight w in C. Then,*

$$A_w = \binom{n}{w} \sum_{i=0}^{w-d} \binom{w}{i} \left( p^{mr(w+1-d-i)} - 1 \right). \tag{17}$$

*2.4. Self-Dual Codes and Building-Up Construction*

Next, we define the usual inner product: for $\mathbf{x}, \mathbf{y} \in GR(p^m, r)^n$,

$$\mathbf{x} \cdot \mathbf{y} = x_1 y_1 + \cdots + x_n y_n. \tag{18}$$

For a code $C$ of length $n$ over $GR(p^m, r)$, let

$$C^\perp = \{ \mathbf{x} \in GR(p^m, r)^n \mid \mathbf{x} \cdot \mathbf{c} = 0, \ \forall \mathbf{c} \in C \} \tag{19}$$

be the dual code of $C$. If $C \subseteq C^\perp$, we say that $C$ is self-orthogonal, and if $C = C^\perp$, then $C$ is self-dual.

Many construction methods are employed for self-dual codes. Among them, the building-up construction method has been extensively used. In this study, we use the method for constructing MDS self-dual codes over Galois rings. In the following theorem, we state the method.

**Theorem 2** ([9]). *Let R be a finite chain ring, let $C_0$ be a self-dual code over R of length n with $k(C_0) = k$, and let $G_0$ be a $k \times n$ generator matrix for $C_0$. Let $a \geq 1$ be an integer and let X be an $a \times n$ matrix over R such that $XX^T = -I$. Let U be an $a \times a$ matrix over R such that $UU^T = -I$, and let 0 be an $a \times a$ zero matrix. Then, the matrix*

$$G = \left( \begin{array}{c|c|c} I & 0 & X \\ \hline -G_0 X^T & G_0 X^T U & G_0 \end{array} \right) \tag{20}$$

*generates a self-dual code C of length n + 2a over R.*

*2.5. MDS Self-Dual Codes*

In this study, we are interested in MDS self-dual codes that are MDS and self-dual. MDS self-dual codes over $GR(2^m, r)$ were constructed using Reed–Solomon codes [4].

**Theorem 3** ([4]). *Let $R = GR(2^m, r), n = 2^r - 1 (> 2)$, and $m \geq 1$. Then, there exists an MDS self-dual code over R with parameters $[2^r, 2^{r-1}, 2^{r-1} + 1]$, which is an extended RS code.*

Kim and Lee investigated the existence of MDS self-dual codes of length $n$ over $GR(p^m, r)$, where $p \equiv 1 \pmod 4$ with any $r$ or $p \equiv -1 \pmod 4$ with even $r$ [5]. The computational results are summarized in Table 1. They constructed many MDS self-dual codes over $GR(p^m, 2)$.

**Table 1.** Existence of MDS self-dual codes of length $n$ over $GR(p^m, 2)$ [5].

| $p$ | $m$ | Length $n$ |
|-----|-----|------------|
| 3 | 2 | $2, 4, 6, 8$ |
| | 3 | $2, 4, 6, 8$ |
| | 4 | $2, 4, 6, 8$ |
| 5 | 2 | $2, 4, 6, 8, 10$ |
| | 3 | $2, 4, 6, 8, 10$ |
| 7 | 2 | $2, 4, 6, 8, 10$ |
| 11 | 2 | $2, 4, 6, 8, 10, 12$ |

The following theorem is very important in the computation of the minimum distance of a linear code over $GR(p^m, r)$.

**Theorem 4** ([16] Corollary 4.3). *If C is a free code over $GR(p^m, r)$, then $d(C) = d(\overline{C})$, where $\overline{C} = \{\overline{c} | c \in C\}$ and $\overline{c}$ is the image of c under the projection of $GR(p^m, r)^n$ onto $GR(p, r)^n$, extended coordinatewise from the projection of $GR(p^m, r)$ to its residue field $GR(p, r)$.*

## 3. MDS Self-Dual Codes over Galois Rings

In the previous section, we saw that MDS self-dual codes over $GR(p^m, r)$ were studied for $p = 2$, and $p \equiv 1 \pmod 4$ with any $r$ or $p \equiv -1 \pmod 4$ with even $r$. However, to the best of our knowledge, MDS self-dual codes over $GR(p^m, r)$ for $p \equiv -1 \pmod 4$ with odd $r$ has not been extensively studied. In this section, we study these codes, and we start with the following theorem.

**Theorem 5.** *Let C be a free self-dual code of length n over $GR(p^m, r)$ for $p \equiv -1 \pmod 4$ with odd r. Then, n should be a multiple of four.*

**Proof.** Let $G$ be a generator matrix of $C$. Because $C$ is free, we assume that

$$G = [I|U] \tag{21}$$

where $I$ is the $\frac{n}{2} \times \frac{n}{2}$ identity matrix and $UU^T = -I$, i.e., $U$ is a $\frac{n}{2} \times \frac{n}{2}$ antiorthogonal matrix. From Table 2, $\frac{n}{2}$ should be even. Therefore, $n$ should be a multiple of four.  □

**Table 2.** Existence of $a \times a$ antiorthogonal matrix $U$ over $GR(p^m, r)$.

| $p$ | $m$ | $r$ | $-1 : SQ$ | $-1 : TSQ$ | Existence of $U$ |
|---|---|---|---|---|---|
| $1 \pmod 4$ | | | Yes | | $\exists \, (a \geq 1)$ |
| $-1 \pmod 4$ | | Even | Yes | | $\exists \, (a \geq 1)$ |
| | | Odd | No | Yes | $\exists \Leftrightarrow a$ is even |
| | 1 | | Yes | | $\exists \, (a \geq 1)$ |
| | | 1 | No | No | $\exists \Leftrightarrow a = 4t \, (t \geq 1)$ |
| 2 | | $2k (k \geq 1)$ | No | Yes | $\exists \Leftrightarrow a$ is even |
| | $\geq 2$ | | | | $a = 2$ or $a$ is odd $\Rightarrow \nexists$ |
| | | $2k + 1 (k \geq 1)$ | No | No | $a = 4t \, (t \geq 1) \Rightarrow \exists$ |
| | | | | | $a = 4t + 2 \, (t \geq 1) \Rightarrow ?$ |

**Corollary 1.** *Let C be an MDS self-dual code of length n over $GR(p^m, r)$ for $p \equiv -1 \pmod 4$ with odd r. Then, n should be a multiple of four.*

**Proof.** From Lemma 1, $C$ is free. Therefore, the result follows from Theorem 5. $\square$

We used the building-up construction in Theorem 2 to construct MDS self-dual codes over $GR(p^m, 3)$ for various $p$s and $m$s of length four and eight, respectively. The computation results are summarized in Table 3. In the following example, we give a detailed explanation for constructing MDS self-dual codes over $GR(3^2, 3)$ of length four and eight.

**Table 3.** Existence of MDS self-dual codes of length $n$ over $GR(p^m, 3)$.

| $p$ | $m$ | Length $n$ |
|---|---|---|
| 3 | 2, 3, 4, 5, 6 | 4, 8 |
| 7 | 2, 3 | 4, 8 |
| 11 | 2 | 4, 8 |
| 19 | 2 | 4, 8 |
| 23 | 2 | 4, 8 |
| 31 | 2 | 4, 8 |

**Example 2.** *Let $C_0$ be a self-dual code of length four over $GR(3^2, 3) = \mathbb{Z}_{3^2}[x]/(f(x))$, where $f(x) = x^3 + 2x + 1$, with generator matrix*

$$G_0 = \begin{pmatrix} 1 & 0 & 2 & 2 \\ 0 & 1 & 7 & 2 \end{pmatrix}. \tag{22}$$

*The minimum weight of $C_0$ is 3, so $C_0$ is an MDS self-dual code. Let*

$$U = \begin{pmatrix} 2 & 2 \\ 7 & 2 \end{pmatrix} \tag{23}$$

*and*

$$X = \begin{pmatrix} 5*w+2 & 6*w^2+3*w+4 & 7*w^2+3*w+6 & 5*w^2+6*w \\ 4*w^2+3*w+6 & 8*w^2+1 & 3*w+5 & 2*w^2+w+1 \end{pmatrix}, \tag{24}$$

where $w = x + (f(x))$. Then, $UU^T = -I$ and $XX^T = -I$. From the building-up construction in Theorem 2, we have

$$G = (G_1 | G_2), \tag{25}$$

where

$$G_1 = \begin{pmatrix} 1 & 0 & 0 & 0 \\ 0 & 1 & 0 & 0 \\ 3*w^2 + 4*w + 4 & w^2 + 7*w & 5*w^2 + 6*w + 1 & w^2 + 5*w + 1 \\ 7*w^2 + 8 & 6*w^2 + 4*w + 7 & 7*w^2 + 8*w + 7 & w^2 + w + 6 \end{pmatrix}, \tag{26}$$

$$G_2 = \begin{pmatrix} 5*w + 2 & 6*w^2 + 3*w + 4 & 7*w^2 + 3*w + 6 & 5*w^2 + 6*w \\ 4*w^2 + 3*w + 6 & 8*w^2 + 1 & 3*w + 5 & 2*w^2 + w + 1 \\ 1 & 0 & 2 & 2 \\ 0 & 1 & 7 & 2 \end{pmatrix}. \tag{27}$$

Then, $G$ generates a self-dual code $C$ of length eight over $GR(3^2, 3)$. The projection of $G$ to its residue field $GR(3, 3)$ is as follows:

$$G_{proj} = \begin{pmatrix} 1 & 0 & 0 & 0 & v^{22} & 1 & v^2 & v^{15} \\ 0 & 1 & 0 & 0 & v^2 & v^{25} & 2 & v^{20} \\ v^9 & v^{10} & v^{25} & v^{18} & 1 & 0 & 2 & 2 \\ v^{12} & v^9 & v^{18} & v^{10} & 0 & 1 & 1 & 2 \end{pmatrix}, \tag{28}$$

where $v = x + (f(x))$ in the residue field $GR(3, 3)$. Let $C_{proj}$ be the code generated by $G_{proj}$ over the residue field $GR(3, 3)$. The minimum weight of $C_{proj}$ is 5, so the minimum weight of $C$ is also 5 based on Theorem 4. Therefore, $C$ is an MDS self-dual code.

The codes in Table 3 were constructed in the same way as in Example 2. In the following, we give $f(x)$, $G_0$, $U$, and $X$ for each code.

1. $p = 3$: $GR(3^m, 3) = \mathbb{Z}_{3^m}[x]/(f(x))$, $f(x) = x^3 + 2x + 1$, $w = x + (f(x))$.

   - $m = 2$

   $$G_0 = \begin{pmatrix} 1 & 0 & 2 & 2 \\ 0 & 1 & 7 & 2 \end{pmatrix}, \quad U = \begin{pmatrix} 2 & 2 \\ 7 & 2 \end{pmatrix}, \tag{29}$$

   $$X = \begin{pmatrix} 5*w + 2 & 6*w^2 + 3*w + 4 & 7*w^2 + 3*w + 6 & 5*w^2 + 6*w \\ 4*w^2 + 3*w + 6 & 8*w^2 + 1 & 3*w + 5 & 2*w^2 + w + 1 \end{pmatrix}. \tag{30}$$

   - $m = 3$

   $$G_0 = \begin{pmatrix} 1 & 0 & 1 & 5 \\ 0 & 1 & 22 & 1 \end{pmatrix}, \quad U = \begin{pmatrix} 1 & 5 \\ 22 & 1 \end{pmatrix}, \tag{31}$$

   $$X = \begin{pmatrix} 17*w^2 + 16*w + 4 & 2*w^2 + 26*w + 26 & 17*w^2 + 12*w + 13 & 26*w^2 + 2*w + 1 \\ 21*w^2 + 4*w + 2 & 18*w^2 + 21*w + 7 & 18*w^2 + 8*w + 11 & 25*w^2 + 4*w + 7 \end{pmatrix}. \tag{32}$$

   - $m = 4$

   $$G_0 = \begin{pmatrix} 1 & 0 & 1 & 22 \\ 0 & 1 & 59 & 1 \end{pmatrix}, \quad U = \begin{pmatrix} 1 & 22 \\ 59 & 1 \end{pmatrix}, \tag{33}$$

   $$X = \begin{pmatrix} 9*w^2 + 50*w + 78 & 42*w^2 + 2*w + 27 & 58*w^2 + 51*w + 73 & 18*w^2 + 20*w + 4 \\ 33*w^2 + 11*w + 52 & 4*w^2 + w + 56 & 6*w^2 + 49*w + 40 & 48*w^2 + 70*w + 14 \end{pmatrix}. \tag{34}$$

   - $m = 5$

   $$G_0 = \begin{pmatrix} 1 & 0 & 1 & 22 \\ 0 & 1 & 221 & 1 \end{pmatrix}, \quad U = \begin{pmatrix} 1 & 22 \\ 221 & 1 \end{pmatrix}, \tag{35}$$

   $$X = \begin{pmatrix} 197*w^2 + 20*w + 71 & 9*w^2 + 191*w + 165 & 133*w^2 + 79*w + 148 & 222*w^2 + 240*w + 223 \\ 206*w^2 + 37*w + 162 & 220*w^2 + 217*w + 21 & 142*w^2 + 40*w + 100 & 122*w^2 + 219*w + 36 \end{pmatrix}. \tag{36}$$

- $m = 6$

$$G_0 = \begin{pmatrix} 1 & 0 & 1 & 221 \\ 0 & 1 & 508 & 1 \end{pmatrix}, \quad U = \begin{pmatrix} 1 & 221 \\ 508 & 1 \end{pmatrix}, \tag{37}$$

$$X = \begin{pmatrix} 532*w^2 + 633*w + 634 & 101*w^2 + 304*w + 98 & 159*w^2 + 328*w + 473 & 691*w^2 + 120*w + 516 \\ 250*w^2 + 92*w + 544 & 379*w^2 + 86*w + 678 & 609*w^2 + 624*w + 203 & 679*w^2 + 223*w + 149 \end{pmatrix}. \tag{38}$$

2. $p = 7$: $GR(7^m, 3) = \mathbb{Z}_{7^m}[x]/(f(x))$, $f(x) = x^3 + 6x^2 + 4$, $w = x + (f(x))$.

- $m = 2$

$$G_0 = \begin{pmatrix} 1 & 0 & 2 & 17 \\ 0 & 1 & 32 & 2 \end{pmatrix}, \quad U = \begin{pmatrix} 2 & 17 \\ 32 & 2 \end{pmatrix}, \tag{39}$$

$$X = \begin{pmatrix} 16*w^2 + 26*w + 31 & 18*w^2 + 16*w + 48 & 39*w^2 + 22*w + 12 & 48*w^2 + 36*w + 14 \\ 37*w^2 + 30*w + 24 & 33*w^2 + 13*w + 22 & 2*w^2 + 47*w + 28 & 33*w^2 + 9*w + 3 \end{pmatrix}. \tag{40}$$

- $m = 3$

$$G_0 = \begin{pmatrix} 1 & 0 & 2 & 32 \\ 0 & 1 & 311 & 2 \end{pmatrix}, \quad U = \begin{pmatrix} 2 & 32 \\ 311 & 2 \end{pmatrix}, \tag{41}$$

$$X = \begin{pmatrix} 88*w^2 + 23*w + 57 & 199*w^2 + 110*w + 293 & 133*w^2 + 221*w + 235 & 212*w^2 + 215*w + 107 \\ 183*w^2 + 321*w + 204 & 215*w^2 + 237*w + 53 & 15*w^2 + 33*w + 147 & 219*w^2 + 148*w + 130 \end{pmatrix}. \tag{42}$$

3. $p = 11$: $GR(11^2, 3) = \mathbb{Z}_{11^2}[x]/(f(x))$, $f(x) = x^3 + 2x + 9$, $w = x + (f(x))$.

$$G_0 = \begin{pmatrix} 1 & 0 & 1 & 19 \\ 0 & 1 & 102 & 1 \end{pmatrix}, \quad U = \begin{pmatrix} 1 & 19 \\ 102 & 1 \end{pmatrix}, \tag{43}$$

$$X = \begin{pmatrix} 87*w^2 + 91*w + 57 & 79*w^2 + 92*w + 40 & 60*w^2 + 83*w + 106 & 20*w^2 + 79*w + 64 \\ 54*w^2 + 68*w + 30 & 84*w^2 + 57*w + 15 & 36*w^2 + 113*w + 31 & 84*w^2 + 37*w + 59 \end{pmatrix}. \tag{44}$$

4. $p = 19$: $GR(19^2, 3) = \mathbb{Z}_{19^2}[x]/(f(x))$, $f(x) = x^3 + 4x + 17$, $w = x + (f(x))$.

$$G_0 = \begin{pmatrix} 1 & 0 & 1 & 63 \\ 0 & 1 & 298 & 1 \end{pmatrix}, \quad U = \begin{pmatrix} 1 & 63 \\ 298 & 1 \end{pmatrix}, \tag{45}$$

$$X = \begin{pmatrix} 234*w^2 + 90*w + 14 & 128*w^2 + 336*w + 104 & 224*w^2 + 255*w + 127 & 174*w^2 + 184*w + 247 \\ 75*w^2 + 211*w + 114 & 51*w^2 + 271*w + 267 & 175*w^2 + 188*w + 56 & 190*w^2 + 168*w + 238 \end{pmatrix}. \tag{46}$$

5. $p = 23$: $GR(23^2, 3) = \mathbb{Z}_{23^2}[x]/(f(x))$, $f(x) = x^3 + 2x + 18$, $w = x + (f(x))$.

$$G_0 = \begin{pmatrix} 1 & 0 & 2 & 169 \\ 0 & 1 & 360 & 2 \end{pmatrix}, \quad U = \begin{pmatrix} 2 & 169 \\ 360 & 2 \end{pmatrix}, \tag{47}$$

$$X = \begin{pmatrix} 126*w^2 + 79*w + 264 & 417*w^2 + 487*w + 466 & 18*w^2 + 280*w + 299 & 480*w^2 + 402*w + 145 \\ 249*w^2 + 180*w + 357 & 143*w^2 + 484*w + 133 & 155*w^2 + 13*w + 23 & 82*w^2 + 44*w + 295 \end{pmatrix}. \tag{48}$$

6. $p = 31$: $GR(31^2, 3) = \mathbb{Z}_{31^2}[x]/(f(x))$, $f(x) = x^3 + x + 28$, $w = x + (f(x))$.

$$G_0 = \begin{pmatrix} 1 & 0 & 4 & 142 \\ 0 & 1 & 819 & 4 \end{pmatrix}, \quad U = \begin{pmatrix} 4 & 142 \\ 819 & 4 \end{pmatrix}, \tag{49}$$

$$X = \begin{pmatrix} 363*w^2 + 303*w + 765 & 333*w^2 + 200*w + 920 & 446*w^2 + 350*w + 211 & 676*w^2 + 391*w + 806 \\ 261*w^2 + 116*w + 339 & 300*w^2 + 108*w + 69 & 373*w^2 + 251*w + 922 & 63*w^2 + 231*w + 597 \end{pmatrix}. \tag{50}$$

## 4. Antiorthogonal Matrices over Galois Rings

In the previous section, we described the use of the building-up construction method [9] to construct MDS self-dual codes. In the construction, it is very important to verify the existence of a square matrix $U$ such that $UU^T = -I$, which is called an antiorthogonal matrix [10,11]. There has been previous research on this problem [12]. In this section, we review this study and further investigate this problem.

*4.1. Review of Previous Research*

We start with the definition of an antiorthogonal matrix. Massey introduced antiorthogonal matrices over finite fields.

**Definition 1** ([10]). *A square matrix $U$ over a finite field $\mathbb{F}$ is said to be antiorthogonal if $UU^T = -I$.*

He characterized self-dual codes using antiorthogonal matrices.

**Theorem 6** ([10]). *Let $C$ be a linear code over a finite field $\mathbb{F}$ with a generator matrix $G = [I|P]$, where $I$ is the identity matrix. Then, $C$ is self-dual if and only if $P$ is antiorthogonal.*

Using the antiorthogonal matrix, he constructed linear codes with complementary duals (LCD codes) [10]. Definition 1 and Theorem 6 can be stated for Galois rings instead of finite fields. Therefore, it is very important to verify the existence of antiorthogonal matrices over Galois rings $GR(p^m, r)$. Han performed a study on the problem [12]. We review the research below.

Suppose that $p \equiv 1 \pmod 4$. Then, $-1$ is a square. In other words, there is an element $\alpha \in GR(p^m, r)$ such that $\alpha^2 = -1$. Let $U = \alpha I$, where $I$ is the $a \times a$ identity matrix for a fixed value $a \geq 1$. Then, $U$ is an antiorthogonal matrix. Therefore, there exists an $a \times a$ antiorthogonal matrix $U$ over $GR(p^m, r)$ for all $a \geq 1$.

Suppose that $p \equiv -1 \pmod 4$. We consider two cases: $r$ is even and $r$ is odd. First, suppose that $r$ is even. Then, $-1$ is a square. From a similar argument to that above, there exists an $a \times a$ antiorthogonal matrix $U$ over $GR(p^m, r)$ for all $a \geq 1$. Second, suppose that $r$ is odd. Then, $-1$ is not a square but a two-square sum. If $U$ is an $a \times a$ antiorthogonal matrix, then $UU^T = -I$. Therefore, $\det(UU^T) = \det(-I)$ and $(\det U)^2 = (-1)^a$. So, $a$ should be even. Because $-1$ is a two-square sum, there exist $\alpha, \beta$ such that $\alpha^2 + \beta^2 = -1$. Let

$$U_2 = \begin{pmatrix} \alpha & \beta \\ \beta & -\alpha \end{pmatrix}. \tag{51}$$

Then, $U_2 U_2^T = -I$. This proves that there is a $2 \times 2$ antiorthogonal matrix $U$. For $a = 2t$, where $t \geq 1$, let

$$U_a = \begin{pmatrix} U_2 & & 0 \\ & \ddots & \\ 0 & & U_2 \end{pmatrix}. \tag{52}$$

Then, $U_a U_a^T = -I$. Therefore, we conclude that there exists an $a \times a$ antiorthogonal matrix $U$ over $GR(p^m, r)$ if and only if $a$ is even.

Suppose that $p = 2$. If $m = 1$, then $-1 = 1$. Therefore, $-1$ is a square. By a similar argument to that above, there exists an $a \times a$ antiorthogonal matrix $U$ over $GR(p^m, r)$ for all $a \geq 1$. Now, suppose that $m \geq 2$. If $r = 1$, then $-1$ is neither a square nor a two-square sum, and it is proven that there exists an $a \times a$ antiorthogonal matrix $U$ over $GR(p^m, r)$ if and only if $a$ is a multiple of four, i.e., $a = 4t, (t \geq 1)$ [19]. Now suppose that $r \geq 2$. We consider two cases: $r$ is even and $r$ is odd. First, suppose that $r$ is even. Then, $-1$ is not a square but is a two-square sum [12]. From a similar argument to that above, there exists an $a \times a$ antiorthogonal matrix $U$ over $GR(p^m, r)$ if and only if $a$ is even.

From the above, the remaining case is that $p = 2$, $m \geq 2$, and $r = 2k + 1 (k \geq 1)$. We know that $-1$ is not a square. Using a similar argument to that above, we know that if $U$ is an $a \times a$ antiorthogonal matrix, then $a$ should be even. Because $\mathbb{Z}_{2^m} \subset GR(p^m, r)$ and there is a $4t \times 4t$ antiorthogonal matrix $U$ over $\mathbb{Z}_{2^m}$, there is an $a \times a$ antiorthogonal matrix $U$ over $R$ for all $a = 4t, (t \geq 1)$. Therefore, our question

is as follows. The first question is "Is $-1$ a two-square sum?" The second question is "Is there an $a \times a$ antiorthogonal matrix $U$ over $GR(p^m, r)$ for $a = 4t + 2, (t \geq 0)$?"

*4.2. Nonexistence of a $2 \times 2$ Antiorthogonal Matrix $U$ over $GR(p^m, r)$, $m \geq 2$ and Odd $r$*

In this subsection, we prove that $-1$ is not a two-square sum in $GR(2^m, r)$ with $m \geq 2$ and odd $r$, and we then conclude that there is no $2 \times 2$ antiorthogonal matrix $U$ over $GR(2^m, r)$ with $m \geq 2$ and odd $r$. We start with the following lemma.

**Lemma 2** ([12]). *If $-1$ is a two-square sum in $GR(2^m, r)$, then $-1$ is a two-square sum in $GR(2^\ell, r)$ for all $1 \leq \ell \leq m$.*

**Corollary 2.** *If $-1$ is not a two-square sum in $GR(2^2, r)$, then $-1$ is not a two-square sum in $GR(2^m, r)$ for all $m \geq 2$.*

The following lemma is the core part of our results.

**Lemma 3.** *Let $r$ be an odd positive integer. Then, $-1$ is not a two-square sum in $GR(2^2, r)$.*

**Proof.** Let $T = \{0, 1, \xi, \xi^2, \ldots, \xi^{2^r-2}\}$ be the Teichmüller representatives in $GR(2^2, r)$ in Section 2. Then, every element $t \in GR(2^2, r)$ can be uniquely represented by the form $t = t_0 + 2t_1$ for some $t_0, t_1 \in T$. Suppose that $-1$ is a two-square sum in $GR(2^2, r)$ and $a^2 + b^2 = -1$ for some $a, b \in GR(2^2, r)$. Let $a = a_0 + 2a_1, b = b_0 + 2b_1$ for some $a_0, a_1, b_0, b_1 \in T$. Then, $a^2 + b^2 = (a_0 + 2a_1)^2 + (b_0 + 2b_1)^2 = a_0^2 + b_0^2$. Therefore $a_0^2 + b_0^2 = -1$. We know that $-1$ is not a square. Therefore, $a_0 \neq 0$ and $b_0 \neq 0$. Thus, $a_0^2 = \xi^j$ and $b_0^2 = \xi^k$ for some $0 \leq j, k \leq 2^r - 2$.

From the above, we have

$$\xi^j + \xi^k = -1. \tag{53}$$

Then, $1 + \xi^j = -\xi^k$. Therefore, $(1 + \xi^j)^2 - \sigma(1 + \xi^j) = (-\xi^k)^2 - \sigma(-\xi^k)$, where $\sigma$ is the automorphism of $GR(2^2, r)$ in Section 2. We have $1 + 2\xi^j + \xi^{2j} - 1 - \xi^{2j} = \xi^{2k} + \xi^{2k}$ and $2\xi^j = 2\xi^{2k}$. Therefore,

$$\xi^j = \xi^{2k}. \tag{54}$$

From Equations (53) and (54), we have $\xi^{2k} + \xi^k = -1$. Therefore, $\xi^k$ is a root of the equation $x^2 + x + 1 = 0$. We applied the map $\overline{\mu}$ in Section 2 to $\xi^k$ and the equation $x^2 + x + 1 = 0$. Then, we have $\overline{\mu}((\xi^k)^2 + (\xi^k) + 1) = 0$ in $\mathbb{F}_{2^r}$. Thus, $(\overline{\mu}(\xi^k))^2 + (\overline{\mu}(\xi^k)) + 1 = 0$ in $\mathbb{F}_{2^r}$. Let $\theta = \overline{\mu}(\xi^k)$. Then, $\theta$ is a root of $x^2 + x + 1 = 0$ in $\mathbb{F}_{2^r}$. Therefore, $\mathbb{F}_{2^r}$ contains the splitting field of $x^2 + x + 1$ over $\mathbb{F}_2$. In other words, $\mathbb{F}_{2^r}$ contains $\mathbb{F}_{2^2}$. This means that $r$ should be even. This leads to a contradiction and the result follows. $\square$

Now, we state the main results of this subsection.

**Theorem 7.** *Let $r$ be an odd positive integer. Then, $-1$ is not a two-square sum in $GR(2^m, r)$ for all $m \geq 2$.*

**Proof.** From Corollary 2 and Lemma 3, we have the result. $\square$

**Theorem 8.** *Let $r$ be an odd positive integer. Then, there is no $2 \times 2$ antiorthogonal matrix over $GR(2^m, r)$ for all $m \geq 2$.*

**Proof.** Suppose that there is a $2 \times 2$ antiorthogonal matrix $U$ over $GR(2^m, r)$. Let

$$U = \begin{pmatrix} a & b \\ c & d \end{pmatrix}. \tag{55}$$

Then, $a^2 + b^2 = -1$. This contradicts Theorem 7. Therefore, we have the result. □

In Table 2, we summarize the results so far for the existence of an $a \times a$ antiorthogonal matrix $U$ over $GR(p^m, r)$. The only remaining problem is for $GR(2^m, r)$, $m \geq 2$, $r = 2k + 1(k \geq 1)$, and $a = 4t + 2(t \geq 1)$. We state this as a research problem as follows.

**Research Problem**: Determine the existence or nonexistence of an $a \times a$ matrix $U$ such that $UU^T = -I$ in $GR(p^m, r)$, where $p = 2$, $m \geq 2$, $r = 2k + 1(k \geq 1)$, and $a = 4t + 2(t \geq 1)$.

**Funding:** This research was funded by the 2019 Professor Education and Research Promotion Program of KoreaTech.

**Acknowledgments:** The author wishes to thank the reviewers for valuable remarks which helped to improve this article.

**Conflicts of Interest:** The author declares no conflict of interest.

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
