# Peer review of "MDS Self-Dual Codes and Antiorthogonal Matrices over Galois Rings"

_information, doi:10.3390/info10040153_

Reviewer 1 Report

The paper is well written and the results are interesting. But I have two major comments:

1) The Introduction has problems with the English language and therefore a moderate editing is needed. For example:

 - the first paragraph sounds strange and must be rewritten;

 - second paragraph, line 3: remove "while they are of interest themselves"

 - for finite fields and for finite rings - over finite ...(over is better)

 - p.1, line -3: ...[9]. In this study, we performedadditional research on this problem. -> ...[9], where we performed additional research on it.

 - Moreover, there are problems with the tenses in the paper. In some paragraphs, present simple, present perfect and past simple are used together. Please read the text carefully and take care of the tenses. 

2)  Theorem 4.2 is not written correctly. You cannot say "nXn linear code". Please write it, as it is in [5].

Author Response

1) The Introduction has problems with the English language and therefore a moderate editing is needed. For example:

 - the first paragraph sounds strange and must be rewritten;

==> Response : I have deleted the first paragraph. I thought that the first paragraph was redundant.

 - second paragraph, line 3: remove "while they are of interest themselves"

==> Response : Done. I have removed that.

 - for finite fields and for finite rings - over finite ...(over is better)

==> Response : Done. I have changed it as the reviewer said.

 - p.1, line -3: ...[9]. In this study, we performed additional research on this problem. -> ...[9], where we performed additional research on it.

==> Response : I think that the original sentences are clearer and better communicates my thoughts. So, I have not changed the original sentences. I hope that the reviewer will understands this.  

 - Moreover, there are problems with the tenses in the paper. In some paragraphs, present simple, present perfect and past simple are used together. Please read the text carefully and take care of the tenses. 

==> Response : I want to note that I examined the paper and also this paper was professionally edited by an English editing company. I hope that the reviewer will understands this.  

2)  Theorem 4.2 is not written correctly. You cannot say "nXn linear code". Please write it, as it is in [5].

==> Response : Done. I modified Theorem 4.2 as the reviewer said.

Reviewer 2 Report

The author studied the self-dual codes over finite rings. Firstly, the author showed constructions

of self-dual MDS codes whose existences were previously unknown

Secondly, the author considered self-dual codes

that are not necessarily MDS. In construction of MDS codes, anti-orthogonal matrices are useful.

the author showed existence and non-existence results of such matrices in terms of the base ring.

The results are clearly written, and I see no serious mistake. I believe that the manuscript

deserves publication.

Author Response

There is no specific suggestion for the revision.

Reviewer 3 Report

The paper is about the construction of MDS self-dual codes over various Galois rings. Overall the paper is well written. However I have a few comments as follows.

In the introduction, it is good to add the original papers of the building-up construction: 

[1] J.-L. Kim, New extremal self-dual codes of lengths 36, 38 and 58, IEEE Trans. Inform. Theory 47 (2001), 386–393.

[2] J.-L. Kim and Y. Lee, An efficient construction of self-dual codes, Bull. Korean Math. Soc. 52 (2015), No. 3, pp. 915–92

In fact GR(p^m, r) with  p= -3 mod 4 and odd r was considered in 

https://arxiv.org/pdf/1201.5689.pdf (J.-L. Kim and Y. Lee, An Efficient Construction of Self-Dual Codes, preprint). So this reference should be also mentioned. On the other hand in this paper, examples have only r=1 case. So the examples in this reviewed paper are new.

Therefore I recommend a revision.

Author Response

==> Response : Done. I have added three references which the reviewer mentioned in the introduction.

Reviewer 4 Report

The paper is well written, with good interest, whereas I cannot accept it in its actual form, it lies deeply on the results of reference [9] in the manuscript.  I suggest to the author to do a major revision, by adding some new results

Author Response

==> Response : I partly agree with the reviewer’s comments. However, in this paper, I constructed MDS self-dual codes over GR(p^m, r) which was not considered in the previous paper. Also, I proved that there is no 2 x 2 antiorthogonal matrix over GR(2^m, r) with m >=2 and odd r. These two things are main difference from the previous paper. The major development which the reviewer suggest will be my next research topic. So, I hope that the reviewer will understand this.

Round  2

Reviewer 1 Report

I don't have more comments.

Reviewer 3 Report

Accept as it is.

Reviewer 4 Report

This version is much better